



# CFC-11 emissions are declining as expected in Western Europe

Alison L. Redington[1,*], Alistair J. Manning[1,*], Stephan Henne[2], Francesco Graziosi[3,4], Luke M. Western[5,6], Jgor Arduini[3], Anita L. Ganesan[7], Christina M. Harth[8], Michela Maione[3], Jens Mühle[8], Simon O'Doherty[5], Joseph Pitt[5], Stefan Reimann[2], Matthew Rigby[5], Peter K. Salameh[8], Peter G. Simmonds[5], T. Gerard Spain[9], Kieran Stanley[5], Martin K. Vollmer[2], Ray F. Weiss[8], and Dickon Young[5]

[1]Met Office Hadley Centre, Exeter EX1 3PB, UK
[2]Empa, Swiss Federal Laboratories for Materials Science and Technology, Dübendorf, Switzerland
[3]Department of Pure and Applied Sciences, University of Urbino, Urbino, Italy
[4]European Commission Joint Research Centre (JRC), Ispra (Va), Italy
[5]School of Chemistry, University of Bristol, Bristol, UK
[6]Global Monitoring Laboratory, National Oceanic and Atmospheric Administration, Boulder, USA
[7]School of Geographical Sciences, University of Bristol
[8]Scripps Institution of Oceanography, University of California San Diego, La Jolla, USA
[9]School of Natural Sciences, University of Galway, Galway, Ireland
[*]These authors contributed equally to this work.

**Correspondence:** Alison Redington (alison.redington@metoffice.gov.uk)

**Abstract.** Production and consumption of CFC-11 (trichlorofluoromethane, $CCl_3F$), CFC-12 (dichlorodifluoromethane, $CCl_2F_2$) and $CCl_4$ (carbon tetrachloride) are controlled under the regulations of the Montreal Protocol and have been phased out globally for dispersive use since 2010. Only $CCl_4$ is still widely produced under exemption as a chemical feedstock (non-dispersive use). After 2010, emissions of CFC-11 and CFC-12 should therefore mostly originate from existing banks (e.g. foams and refrigerators), however evidence emerged of an increase in global emissions of CFC-11, which was in part attributed to eastern China. Emissions of CFC-11, CFC-12 and $CCl_4$ have subsequently declined in this region, however the total global increase in CFC-11 was not fully accounted for. The motivation for this work was to assess the emissions of CFC-11 and the associated gases, CFC-12 and $CCl_4$, from Western Europe. All countries in this region have been subject to the controls of the Montreal Protocol since the late 1980s, and, as non-Article-5 Parties, have been prohibited from producing CFCs and $CCl_4$ for dispersive use since 1995. Four different inverse modelling systems are used to estimate emissions of these gases from 2008-2021 using data from four atmospheric measurement stations: Mace Head (Ireland), Jungfraujoch (Switzerland), Monte Cimone (Italy) and Tacolneston (UK). The average of the four model studies found that Western European emissions of CFC-11, CFC-12 and $CCl_4$ between 2008 and 2021 were declining at 3.5 (2.7-4.8)%, 7.7 (6.3-8.0)% and 4.4 (2.6-6.4)% $yr^{-1}$ respectively. Throughout this period, the highest CFC-11 emissions were in Northern France and Benelux (Belgium, the Netherlands and Luxembourg). Emissions of CFC-12 co-located in this region were slightly higher than elsewhere in Western Europe, and also showed some enhancement of $CCl_4$ emissions. However for $CCl_4$, emissions were highest in the south of France. France had the highest emissions of CFC-11, CFC-12 and $CCl_4$ over the period 2008-2021. Emissions from Western Europe (2008-2021) were on average 2.4 ± 0.4 Gg (CFC-11), 1.3 ± 0.3 Gg (CFC-12), 0.9 ± 0.2 Gg ($CCl_4$). This study concludes that the emis-





sions of CFC-11 from Northern France and Benelux are unlikely to be the result of new production. Our estimated decline in
emissions of CFC-11 is consistent with a Western European bank release rate of 3.4 (2.6-4.5)%, which is in the upper half of
the published range.

**1   Introduction**

Trichlorofluoromethane (CFC-11 or $CFCl_3$), dichlorodifluoromethane (CFC-12 or $CF_2Cl_2$) and carbon tetrachloride ($CCl_4$)
are damaging to the stratospheric ozone layer (Karpechko et al., 2018) and are strong greenhouse gases (Masson-Delmotte
et al., 2021). Their production, consumption and use are controlled through the Montreal Protocol on Substances that Deplete
the Ozone Layer (MP) and its amendments. For non-Annex 5 parties, including all of the countries in Western Europe, the
production and consumption of CFC-11 and CFC-12 for dispersive uses have been banned since 1995. For developing countries
(Annex-5), production and consumption of CFC-11 and CFC-12 have been banned since 2010. CFC-11 was mainly used in
aerosol spray cans, as a solvent and as an agent for blowing foams into buildings and consumer products; CFC-12 was mainly
used in refrigerators, air conditioning units and as a foam blowing agent. $CCl_4$ was used historically as a solvent and also as a
feedstock to produce other chemicals, predominantly CFC-11 and CFC-12. Production and consumption of $CCl_4$ for dispersive
processes has been banned under the MP since 2010. However, non-dispersive uses (such as a chemical feedstock) are exempt
from the MP and it is permitted to be produced under the assumption that the majority of $CCl_4$ made, is fully converted into
the target chemical, recycled or destroyed.

    Globally, emissions of CFC-11 and CFC-12 have been decreasing as a result of the MP. The CFC that remains in products
and equipment is described as a bank, which is characterised as active if the product is still in use or as inactive if the product
has been decommissioned. Nearly all the active bank is comprised of foam panels and boardstock in buildings. Nearly all the
remaining bank of foams used in refrigerating applications have been decommissioned and either landfilled or destroyed. The
European Union requires ozone-depleting blowing agents to be captured and destroyed under the EU Directive 2002/96/EC
(EU, 2003). TEAP (2019) analysis suggests that globally there are an estimated $750 \pm 50$ Gg of CFC-11 in active foam banks
and $700 \pm 50$ Gg in inactive banks in 2021. The majority of the global CFC-11 bank is in North America and Europe.

Emissions of $CCl_4$ have also been decreasing globally since the 1990's as a result of the MP, however there is a gap between
expected emissions and those calculated by inverse modelling techniques and atmospheric measurements, as highlighted in
Carpenter et al. (2014) and Liang et al. (2014). The global rate of decrease in the mole fraction of CFC-11 had slowed from
2013 onward (Montzka et al., 2018) and the most likely cause was an increase in emissions of CFC-11 from eastern Asia. This
finding was supported by Rigby et al. (2019), who identified emissions in eastern China ($7.0 \pm 3.0$ Gg yr$^{-1}$ 2014-2017) that



explained 40-60% of the increased global emissions. In 2019, CFC-11 emissions from eastern China rapidly declined and they are now similar to before this period of renewed production and use (Park et al., 2021; Montzka et al., 2021). CFC-12 and $CCl_4$ emissions are often associated with production of CFC-11, and emission levels were shown to be higher than expected in eastern China after 2013. They subsequently decreased just before the reduction in CFC-11 (Park et al., 2021). Lunt et al. (2018) reported a lack of decline in $CCl_4$ emissions during the period 2009-2016 in eastern China, also suggesting globally

significant $CCl_4$ sources in this region.

Hu et al. (2022) used observational data from two global aircraft surveys to assess the continental scale contributions to global CFC-11 emissions from 2012-2017 to try to understand the additional increase in global CFC-11 not accounted for by the increased emissions found in Eastern China (Rigby et al., 2019). They report that in addition to eastern mainland China's contribution, emissions likely came from temperate western Asia and tropical Asia.

Fraser et al. (2020) present a comprehensive review of CFC emissions in Australia from 1960 to 2017, where they conclude that Australian emissions since the early 1990's were declining at the same rate as the global emissions; around 10% per year. They found no evidence of renewed emission or consumption of CFCs in Australia. $CCl_4$ emissions in Australia were assessed by Fraser et al. (2014) from 1996 to 2011, where they identified potential local emission sources associated with contaminated soils, toxic waste treatment facilities and chlor-alkali plants. They concluded that currently unaccounted-for emissions in other

regions could arise from similar sources.

Hu et al. (2016) reported on continued emissions of $CCl_4$ in the United States (2008-2012) which are nearly two orders of magnitude higher than those estimated by the US Environmental Protection Agency inventory for industrial chemical processes.

CFC-11, CFC-12 and $CCl_4$ emissions from Europe have been reported in previous studies. Manning et al. (2003) estimated emissions of CFC-11 and CFC-12 from 1995 – 2002 for Western Europe as 8.9 and 14.3 $\mathrm{Gg\,y}^{-1}$, respectively. Keller et al.

(2011) used measurements at Mace Head, Jungfraujoch and K-Puszta (Hungary) to estimate emissions of CFC-11 of 2.1 $\pm$ 0.4 $\mathrm{Gg\,y}^{-1}$ in 2009 from North-West Europe (Ireland, UK, France, Benelux, Germany and Denmark) and 4.2 $\pm$ 1.2 $\mathrm{Gg\,y}^{-1}$ in the whole of Europe (excluding Norway, Sweden, Finland and Russia). In the same study, CFC-12 emissions of 1.0 $\pm$ 0.4 $\mathrm{Gg\,y}^{-1}$ in North-West Europe and 2.2 $\pm$ 1.1 $\mathrm{Gg\,y}^{-1}$ in the whole of Europe were estimated. Graziosi et al. (2016) reported on European $CCl_4$ emissions over the period 2006 to 2014. They found that France was the main source of emissions. On average

European emissions contributed 4.0% of global emissions for 2006-2012.

Here we use four inverse modelling systems, employing two atmospheric transport models with different meteorological inputs and different inversion approaches to asses CFC-11, CFC-12 and $CCl_4$ emissions from western Europe from 1990 - 2021 and their rates of decline. We identify whether these rates are consistent with what is expected from the consumption and production controls imposed by the MP and present spatial distributions of the emission estimates, averaged in time and over

the four model systems.

Section 2 describes the atmospheric measurements and inverse frameworks used, section 3 presents the emission estimates and their discussion and finally concluding remarks are made in section 4.



## 2 Methods

### 2.1 Atmospheric Measurements

We use the in-situ high-frequency observations from four atmospheric monitoring stations which are part of the Advanced Global Atmospheric Gases Experiment (AGAGE) network (Prinn et al., 2018): Mace Head (MHD) on the west coast of Ireland, Jungfraujoch (JFJ) in the Swiss Alps, Monte Cimone (CMN) in the northern Apennine Mountains in Italy and Tacolneston (TAC) in the south-east of the UK (see Table 1). TAC and MHD are also part of the UK DECC network (Stanley et al., 2018). Data from all four stations are routinely inter-compared to ensure good data quality.

The measurements at TAC and JFJ were made using Medusa gas chromatograph (GC) mass spectrometer (MS) instruments (Miller et al., 2008). Typically, 2-litre samples of ambient air are pre-concentrated on a cold trap, then cryo-distilled and cryo-focussed on a second cold trap held at ∼-160°C, where the main constituents in air (oxygen, nitrogen, noble gases, carbon dioxide and water) are removed. The sample is then transferred to the GC with subsequent quadrupole MS detection. A sample run typically takes 1 hour, and air sample measurements are bracketed by calibration standard measurements to correct for 95 MS drift and to quantify the trace gases in the air samples, thus 2-hourly atmospheric observations are available from these systems. Typical measurement precisions for CFC-11, CFC-12 and $CCl_4$ from these instruments are ∼0.22% (1 sigma std) ∼0.14% and ∼1.11% respectively.

The measurements at MHD from 1994 onwards were made with a GC Multiple Detector (MD) (Simmonds et al., 1995) system. These measurements are made every 20 minutes and using bracketing standard measurements result in an ambient air 100 sample observation every 40 minutes. Precision for these measurements (∼0.10%, ∼0.08% and ∼0.32% for CFC-11, CFC-12 and $CCl_4$ respectively) are generally better than that achieved using the Medusa-GC-MS system.

Measurements at CMN started in Feb 2006. The system consists of a commercial Thermal Desorption – GC-MS system (Markes International Unity2-AirServer2) coupled with a GC-MS (Agilent GC 6850 MS5975C) (Maione et al., 2013) to enrich halocarbons on the adsorbing trap; ambient air samples are collected every second hour and bracketed with working 105 standard runs following the AGAGE-Medusa protocol. The measurement precisions are ∼0.40%, ∼0.31% and ∼0.47% for CFC-11, CFC-12 and $CCl_4$ respectively.

All measurements are based on Scripps Institution of Oceanography (SIO) primary calibration scales SIO-05. The accuracies of these calibration scales are estimated at 2%. The calibration scales are propagated to the field instruments using travelling tertiary whole-air standards exchanged between SIO and the field sites (for CMN, calibrated standards are exchanged with 110 MHD). Tertiary standards are used on site for the calibration of quaternary whole-air standards, which are used to bracket the air measurements. This propagation of standards adds a statistically independent uncertainty of 1 – 2% to the measurements. Overall, the accuracy of the air measurements is ∼3% and includes calibration scale, propagation and reproducibility of the air measurements. For more details see, for example, Vollmer et al. (2016).



**Table 1.** Observation site information. Note the site altitude is given in metres above sea level (m asl) and inlet height as metres above ground (m agl)

| Site Name | Country | Site Acronym | Latitude (°N) | Longitude (°E) | Site altitude (m asl) | inlet height height (m agl) | Dates obs. available |
|---|---|---|---|---|---|---|---|
| Mace Head | Ireland | MHD | 53.33 | -9.904 | 8 | 10 | 1994–2021 |
| Tacolneston | UK | TAC | 52.52 | 1.139 | 56 | 100 | 2012–2017 |
| Tacolneston | UK | TAC | 52.52 | 1.139 | 56 | 185 | 2017–2021 |
| Jungfraujoch | Switzerland | JFJ | 46.55 | 7.986 | 3580 | 10 | 2008–2021 |
| Monte Cimone | Italy | CMN | 44.18 | 10.70 | 2165 | 8 | 2006–2021 |

## 2.2 Inversion Frameworks

115 Four different atmospheric inverse modelling systems were employed to estimate emissions of CFC-11, CFC-12 and $CCl_4$ from Western Europe (WEU) where WEU comprises of Ireland, the UK, France, Belgium, the Netherlands, Luxembourg, Germany, Italy, Switzerland, Austria and Denmark. Each system consists of an atmospheric transport model (ATM) and a Bayesian optimisation framework that infers emission into the atmosphere from the mole fractions observed at the measurement sites. All of the systems have been applied in previous studies and are briefly described here.

120 Both of the ATMs used are Lagrangian Particle Dispersion Models (LPDMs) that use 3-dimensional modelled meteorology from operational meteorological centres. They were both run backward in time, releasing model particles at the location of the atmospheric observations and thereby calculating source receptor relationships (SRR, also referred to as source sensitivities), which are required for the inverse estimate of emissions (Seibert and Frank, 2004). The SRRs represent the quantitative link between an emission source at any location in the model domain and the change in mole fraction at a measurement site. As 125 the main sources are expected to be close to the ground, we evaluate the SRRs from near to the ground (for example 0-40m a.g.l. in NAME) to the sampling location and height. An overview of the ATMs and the release settings used are given in the following sections and Table 2. Two common a priori emission fields were used by each model to test the sensitivity of the inverse results; the first, 'flat' uniform emissions across the land area of Europe, and the second, population-weighted emissions over land areas. CFC-11 and CFC-12 were 3.0 kg km$^{-2}$ yr$^{-1}$ for the flat land a priori emissions which resulted in 130 6.6 Kt yr$^{-1}$ over WEU. $CCl_4$ was emitted at 1.2 kg km$^{-2}$ yr$^{-1}$ for the flat land a priori emissions which gave 2.6 Kt yr$^{-1}$ over WEU. The population-weighted a priori emissions used the same totals for WEU. A common set of quality controlled observational data (Section 2.1) including instrument precision, was compiled from the AGAGE database and shared among all modelling groups. From these observations each modelling team estimated background mole fractions following their method of choice; this enables calculation of enhancements above baseline. Individual groups used different temporal aggregation of 135 the observational data to best fit their modelling frameworks.




**Table 2.** The inversion systems: ATMs, meteorology, geographical domains over which the ATMs are run, number of particles released per hour and the inversion time-steps.

| Inversion System | Atmospheric Transport Model | Driving Meteorology | Computational Domain | Inversion Domain | Particles Released (hr$^{-1}$) | Inversion Time Step |
|---|---|---|---|---|---|---|
| InTEM | NAME | Unified Model | -98.1 to 39.6°E, 10.6 to 79.2°N | -14.3 to 30.8°E, 36.4 to 66.3°N | 20,000 | 4 hr |
| Empa | FLEXPART 9.1 | ECMWF-IFS operational | Global Alpine nest | -12.0 to 26.4°E, 36.0 to 62.0°N | 16,667 | 3 hr |
| University Urbino | FLEXPART 10.4 | ECMWF operational | Global | -20.0 to 50.0°E, 0.0 to 80.0°N | 13,333 | 3 hr |
| Bristol-MCMC | NAME | Unified Model | -98.1 to 39.6°E, 10.6 to 79.2°N | -98.1 to 39.6°E, 10.6 to 79.2°N | 20,000 | 12 hr |

### 2.2.1 InTEM

The Inverse Technique for Emission Modelling (InTEM) (Arnold et al. (2018), Manning et al. (2021)) is used by the UK government to verify its nationally reported greenhouse gas emissions. InTEM uses the Numerical Atmospheric dispersion Modelling Environment (NAME, Jones et al., 2007) LPDM which has been used for many similar studies (Manning et al., 2011; Say et al., 2016; Lunt et al., 2018; Fraser et al., 2020; Say et al., 2020b; Ganesan et al., 2020; Rigby et al., 2019; Park et al., 2021). NAME is driven by three-dimensional meteorology; the horizontal and vertical resolution of the meteorology has increased over the modelled period (see Manning et al. (2021)). The NAME model simulates atmospheric dispersion by the release of thousands of particles into the modelled atmosphere, here, 20,000 particles per hour from each station (at heights of 10 m agl for MHD, 100 and 185 m agl for TAC, 1000 m agl for JFJ, and 500 m agl for CMN), and followed backwards in time for 30 days or until they leave the computational domain. InTEM is a Bayesian system that minimises the mismatch between the model and the atmospheric observations given the constraints imposed by the observation and model uncertainties and prior information with its associated uncertainties. The 3-D varying background mole fraction and observation station bias are solved-for within the inverse system along with the spatial distribution and magnitude of the emissions. A time-varying prior background mole fraction is derived from the MHD, JFJ and CMN observations as described in Manning et al. (2021). TAC uses the same prior background mole fraction as used at MHD. The prior bias for each station is set to zero with an uncertainty of 1.2 ppt for CFC-11 and CFC-12 and 0.43 ppt for CCl$_4$ (see Manning et al. (2021)). The InTEM inversions used a prior uncertainty of 500% over WEU. The observations are averaged into 4-hr periods. The uncertainty of the observations is derived from the reported daily observation precision uncertainty and the variability of the observations within a 12-hr period. The modelling uncertainty for each 4-hr period is the larger of the median pollution (above baseline) event in that year, or 10% of the magnitude of the pollution event.

### 2.2.2 Empa

The Empa (Swiss Federal Laboratories for Materials Science and Technology) Bayesian inverse modelling framework (Henne et al. (2016)) has been used e.g., to estimate methane emissions in Switzerland and halocarbon emissions in Europe (e.g., Brunner et al., 2017; Schoenenberger et al., 2018; Simmonds et al., 2020) and eastern Asia (Rigby et al., 2019; Park et al., 2021).



The system uses source sensitivities as calculated by the LPDM FLEXPART (Version 9.1_Empa, Stohl et al. (2005); Pisso et al. (2019), http://www.flexpart.eu) driven by operational meteorological analysis fields provided by the ECMWF-IFS model extracted at a global resolution of 1°x1° and a higher resolution nest (0.2°x0.2°) over the Alpine area. FLEXPART was run in backward mode, releasing 50,000 particles every 3 hours at each receptor site and following these 10 days backward in time. For elevated sites, model particles were released at a suitable altitude above model ground that is between model topography and real site altitude (Keller et al., 2011). Hence, 3000 m and 2000 m above sea level were chosen for JFJ and CMN, respectively. SRRs were evaluated from the surface to 100 m above ground.

The Bayesian inversion was carried out to estimate annual mean emissions on an irregularly-sized grid covering Western Europe. Grid sizes were inversely proportional to SRRs, resulting in generally finer grid resolution close to the observational sites. The employed covariance matrices consider correlated uncertainties in space in the emission a priori and correlated uncertainties in time for the data-mismatch. The inversion framework offers different options for choosing uncertainty parameters. Here, we chose a semi-objective approach setting the total uncertainty of the prior emissions to 100% for the whole inversion domain with a correlation length scale of $100 \, \mathrm{km}$ and an iterative approach for estimating the data-mismatch uncertainty from the model-observation residuals assuming a linear relationship between data-mismatch uncertainty and simulated total source sensitivity (Henne et al., 2016). The temporal correlation coefficient of the data-mismatch covariance was estimated from an exponential fit to the auto-correlation function of the model residuals.

Baseline concentrations were estimated for each site using a statistical filter (REBS) on the observations (Ruckstuhl et al., 2012). For each site the baseline was incorporated as a linear interpolation between baseline nodes. Baseline concentrations at the nodes were part of the state vector and optimised along with the emissions. Baseline nodes were spaced 14-days apart. Baseline uncertainties were taken from the REBS filter and assumed to be correlated with a time scale of 30 days.

The appropriate selection of uncertainty parameters was tested by evaluating the reduced $\chi^2$ index of the covariance matrices. $\chi^2$ values were in the range of 0.85 to 1.15, 0.65 to 0.9, and 0.7 to 0.95 for CFC-11, CFC-12 and CCl$_4$, respectively, indicating a balanced setup.

### 2.2.3 University Urbino

The University of Urbino inversion system used here is very similar to that described in (Stohl et al., 2009). The inversion system has already been applied to estimate CCl$_4$ emissions, Graziosi et al. (2016), and greenhouse gases in the European domain Simmonds et al. (2020). Like the Empa system, the method is based on transport simulations using the FLEXPART (Version 10.4, Pisso et al. (2019)) LPDM. However, FLEXPART was run with a different configuration and different input data. Here, FLEXPART was used to simulate the dispersion of 40,000 particles, released from each receptor (measurement) site, every three hours, and followed for 20 days backward in time, calculating source sensitivities from the ground to 100 m above ground. FLEXPART was driven by operational 3-hourly meteorological data from ECMWF at 1° ×1° latitude and longitude resolution, from 2008 to 2021.

The University of Urbino inversion system is based on the analytical inversion method of Seibert (2001), which was subsequently developed and evaluated by Stohl et al. (2009). In this implementation we use the same mole fraction background



filtering method, optimized by the inversion scheme as described in this paper. With the purpose of reducing the number of
unknowns in the inversion procedure, a variable resolution grid is created, having a higher (lower) resolution over areas associ-
ated with high (low) sensitivity to the observation sites. Uncertainty in the inversion grid is set to 250% of prior emission flux
in each gridbox, for both the flat and population priors. In general, larger uncertainty reduction is associated with areas with
larger source sensitivity, and lower uncertainty reduction is associated with lower source sensitivities.

### 2.2.4   Bristol-MCMC

The Bristol-MCMC inverse system follows Say et al. (2020a) and Western et al. (2021) with minor modifications. The sensitiv-
ities of the measurements to emissions were derived using NAME driven by meteorology from the Met Office Unified Model
global product (Walters et al., 2014), not using the variable UK high resolution (UKV) component as used in InTEM. The
spatial resolution of the global product varied over the period of study (as detailed in Manning et al. (2021)) and has a 3-hourly
temporal resolution. The total model domain is detailed in Table 2, and sensitivities are output at a horizontal resolution of
$0.352°$ (lon) by $0.234°$ (lat) degrees.

Uncertainties in a priori emission estimates, the background (i.e. the mole fraction contribution outside the model domain)
and model transport are estimated following Say et al. (2020a). This uses a Markov chain Monte Carlo (MCMC) algorithm
to estimate the joint-posterior distribution (Ganesan et al., 2014), where the reported values are the posterior mean and 68%
highest posterior density region (see Western et al., 2021). Inference is carried out as a scaling of the a priori emissions
and boundary mole fraction, where the latter was taken from the AGAGE 12-box model (Cunnold et al., 1983; Rigby et al.,
2013, 2014; Montzka et al., 2021). The boundary mole fractions are inferred using a single value for each month at each
model boundary, with an additional bias term (additive offset) between each site for each year to allow for small systematic
differences in the modelling and measurements. This bias term is in reference to measurements at Mace Head. The likelihood
error contains three components added together in quadrature. The first is the measurement error, second is a term equal to 10%
of the above-background mole fraction and the third is an additional model error term, which is estimated during inference.
The emissions are divided into 199 basis functions, chosen using a quadtree algorithm (Finkel and Bentley, 1974) based on the
a priori contribution to the mole fraction.

Prior uncertainty was set using a truncated Normal distribution (with a lower bound at 0 to prevent negative emissions) and
a standard deviation of 5. The prior distribution for the bias term was Normally distributed with a mean of zero and a standard
deviation of 1 ppt. The prior for the model error followed a log-normal distribution with $\mu = 1.5$ and $\sigma = 0.5$ and the boundary
condition followed a log-normal distribution with $\mu = 0.004$ and $\sigma = 0.02$.

### 3   Results & Discussion

Estimating the background mole fractions at each measurement station is an integral part of each inversion system. Figure 1
shows example estimates from the InTEM system for CFC-11, CFC-12 and $CCl_4$ at MHD from 1990. The monthly Northern
Hemisphere background trend is derived from MHD observations using the UK Met Office background methodology (Manning





et al., 2021). The average rate of decline of the Northern Hemisphere mole fractions of CFC-11 at MHD between 2008 and 2021 is 1.7 ppt yr$^{-1}$, but it has not been uniform as discussed in (Montzka et al., 2018). Between 2008 and 2012 at MHD it was 1.9 ppt yr$^{-1}$ and between 2013 and 2018 it was 1.2 ppt $^{-1}$. The slowdown in the decline has been attributed in part to
enhanced emissions of CFC-11 from China (Rigby et al., 2019). The average rates of decline of the Northern Hemisphere mole fractions of CFC-12 and CCl$_4$ at MHD between 2008 and 2021 are 3.2 and 1.0 ppt yr$^{-1}$, respectively.

Each inverse modelling system employs a priori background mole fractions estimated using different techniques and adjusts these backgrounds within the inverse systems to produce a posterior background. A comparison of the posterior backgrounds at MHD from the four inverse systems is shown in the supplementary Figure S1. The range of the models offset from the
four-model average is generally within ± 0.25 ppt for all three gases, with the InTEM model having a positive offset for all 3 gases, MCMC mostly positive for CFC-11 and CFC-12 with very little offset for CCl$_4$, Empa having largely negatives offsets from the mean for all gases and Urbino having largely negative offsets from the mean for CFC-11 and CFC-12 but very little offset for CCl$_4$.

Analysis of the MHD observational records for the three gases from 1989 has been undertaken to assess how the number of
pollution peaks (i.e. excursions from background) has changed over time. Figure S2 shows the percentage of observations per year greater than the larger of the standard deviation of the InTEM background and twice the instrument precision, per year for each gas. The decrease in this percentage over the time period, for all three gases, clearly demonstrates that the emissions of these gases from WEU have fallen significantly over this period.

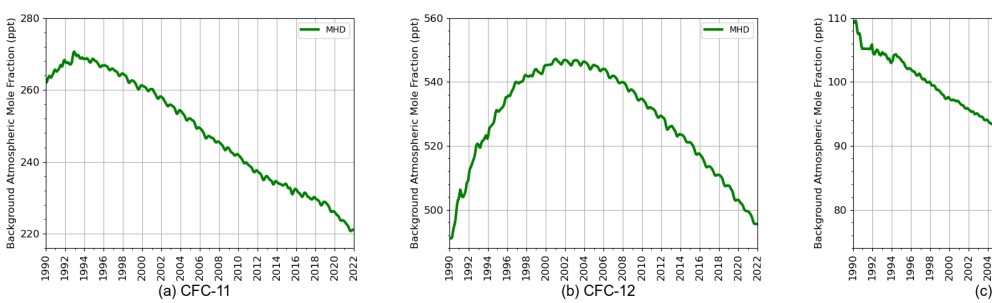

**Figure 1.** Background mole fractions estimated from the observations at Mace Head (Ireland) for (a) CFC-11, (b) CFC-12 and (c) CCl$_4$

The average yearly geographical sensitivities to surface emissions (footprints) at the four observation sites for the different
LPDMs for the period 2017-2021 are shown in Figure 2. Areas of high sensitivity are seen close to the measurement stations, but the footprints show that emissions from countries such as Spain and Poland are unlikely to be significant at the stations and hence are not included in our definition of WEU.





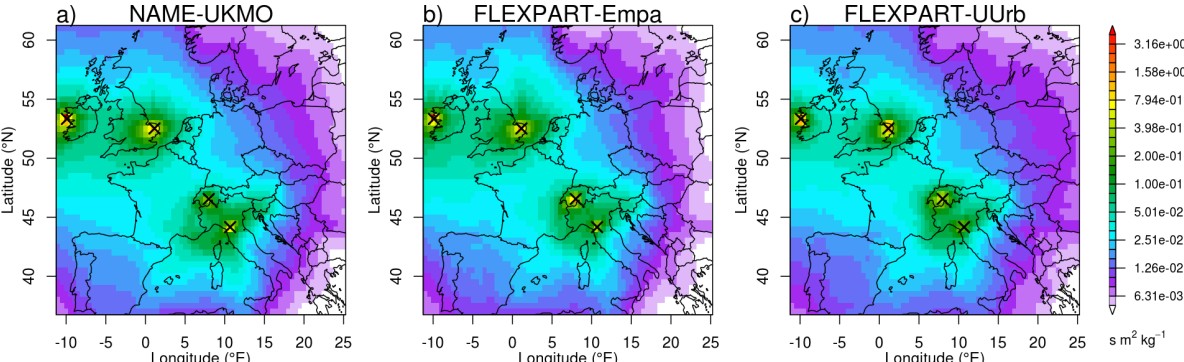

**Figure 2.** Average model sensitivity to European surface emissions at the four observation stations (marked as x) for the period 2017-2021 for three of the atmospheric model configurations. The footprint used by Bristol-MCMC is indistinguishable from panel (a) and is not shown.

Figure 3 shows the WEU emissions (2008-2021) estimated by the four different modelling systems, and the average of the four in black, with associated uncertainty estimates for CFC-11, CFC-12 and CCl$_4$ in the left hand panels, and InTEM-only emission estimates for the three gases from 1990-2021 in the right hand panels. The WEU emission estimate results in Figure 3, and throughout the paper, are from the flat prior case, with the results from the population prior case shown in the supplement. From 1990 until 2008, only observations from MHD were available, which greatly limits the ability to estimate emissions over WEU due to the reduced footprint. To increase the number of observations per inversion, the inversion time period was increased to 2-years, advanced in steps of 1-year. Additional observations were included as they became available (CMN in 2006, JFJ in 2008 and TAC in 2012). The annual emissions shown are the average of the two inversions that include that year, with the exception of 1990 and 2021 which are based on a single two year inversion; there are still large uncertainties in the early years, as seen in panels (b), (d) and (f). The results in Figure 3 panels (b) and (d) show that WEU emissions of CFC-11 and CFC-12 declined sharply between 1990 and 1994 prior to the phase-out in Europe in 1995. Since then there has been a slow decline, likely relating to the gradual release from CFC-11 and CFC-12 "banks", which is the amount of each gas held in existing infrastructure. The estimated emissions of CCl$_4$ also decreased rapidly from 1990 to 1993, but increased again in 1995-1996 before declining to the current day.

The main focus of the study is the period from 2008 to 2021, when measurement data are available from at least three European sites as shown in 3 panels (a), (c) and (e) for CFC-11, CFC-12 and CCl$_4$, respectively. Panel (a) shows the estimated CFC-11 emissions from WEU and the four modelling systems are in good agreement, with InTEM and Empa generally estimating lower values and Urbino and Bristol-MCMC higher values. The model average trend is downwards, from $3.1 \pm 0.6$ Gg yr$^{-1}$ in 2008 to $1.7 \pm 0.2$ Gg yr$^{-1}$ in 2021, although there is some variability. The average decrease per year is 3.5 (2.7-4.8)% (gradient of the four model average compared to the average value; range of the model gradients). Similarly there is an overall downward trend in CFC-12, from $2.2 \pm 0.4$ Gg yr$^{-1}$ in 2008 to $0.9 \pm 0.2$ Gg yr$^{-1}$ in 2021. Again there is both year-to-year and model-to-model variability, with the InTEM model generally estimating lower than the average values and the Bristol-MCMC model estimating higher than the average values. The average decrease per year is 7.7 (6.3-8.0)%. Panel



(e) shows a flatter trend for $CCl_4$ with the InTEM model being more of an outlier with consistently lower estimates compared with the other three. The model average WEU emissions are $1.3 \pm 0.3$ Gg yr$^{-1}$ in 2008 declining to $0.6 \pm 0.1$ Gg yr$^{-1}$ in 2021. The average decrease per year is 4.4 (2.6-6.4)% for $CCl_4$.





**Figure 3.** Panels (a), (c) and (e) show estimated annual emissions from 2008 to 2021 for the four models, with 1 standard deviation uncertainty (shading), for WEU, with the average shown in black, with uncertainty the average of the four models (grey shading), for CFC-11, CFC-12 and CCl$_4$ respectively. Panels (b), (d) and (f) show the annualised, 2-year InTEM inversion emission estimates from 1990 to 2021 for WEU, for CFC-11, CFC-12 and CCl$_4$ respectively.





Figure 4 shows the four model average annual posterior CFC-11, CFC-12 and CCl$_4$ emission distributions (2013 to 2021)
estimated using the flat prior distribution. As each inversion system used a different inversion grid, the solutions were re-
sampled to a standard grid and averaged. The models are consistent in their emission distribution, as shown quantitatively by
the relative standard deviations in panels (b), (d) and (f). The model results for flat and population prior inversions are compared
in the supplement (Figures S3, S4, S5, S6, S7 and S8).

The average emission distribution, Figure 4 panel (a), indicates elevated emissions of CFC-11 over Belgium, southern
Netherlands, northern France and west Germany. A more precise location is not possible given the data sparsity, the precision
of the observation network and the uncertainty of the atmospheric modelling. The average emission distribution of CFC-12,
Figure 4 panel (c), is largely uniform over land, with slight enhancements in Belgium, southern Netherlands, west Germany
and the far south east of France. Panel (e) shows the average emission distribution of CCl$_4$, which indicates that the largest
emissions are in the south-east of France and some elevated emissions in the east of France, Belgium and central UK.

We evaluated the models' ability to reproduce the observed concentrations at the four observational sites (2013-2021),
focusing on the regional signal (observation minus baseline) only, i.e. the ability of the models to reproduce pollution events
with the correct magnitude. Taylor diagrams of this analysis are available in the supplement (Figure S9). We observe a large
spread in prior model performance with considerable differences across the four model systems. Performance significantly
improved in the posterior solutions, with the different systems much more aligned, no one model stands out. Performance was
generally better for the low-altitude sites (MHD and TAC), both in terms of correlation and normalised standard deviation,
compared to the high altitude sites, reflecting the challenges of atmospheric transport simulations to such locations.







**Figure 4.** Posterior (a) CFC-11, (c) CFC-12 and (e) CCl$_4$ emission distributions averaged over the period 2013 to 2021 and over the four inversion systems using the flat prior. Relative standard deviations of CFC-11, CFC-12 and CCl$_4$ emissions over the four inversion systems are shown in (b), (d) and (f).



Western Europe (WEU) has been further divided down to country level and the averages of the four models are shown in Figure 5. Panels (a), (c) and (e) show emissions per year per country in Gg per year. Panels (b), (d) and (f) show population averaged emission from 2008 to 2021 in Gg per year per million of population. Austria and Denmark are included in the WEU total but not presented separately due to their large individual emission uncertainties resulting from country size and location with respect to the observation stations. Benelux represents the combined emissions from Belgium, the Netherlands and Luxembourg.

Panel (a) shows that CFC-11 emissions from Ireland and Switzerland are very low (<0.1 Gg per year); emissions from Benelux, the UK, Italy and Germany range from 0.1 to 0.6 Gg per year; and France has emissions higher than any other country with more than 0.7 Gg per year until 2018, dropping to 0.5 - 0.6 Gg per year thereafter. Panel (b) shows that France also has the highest emissions per capita, followed by Benelux countries.

Figure 5 panel (c) shows that CFC-12 emissions by country between 2008-2021 declined more consistently than CFC-11 emissions. The distribution of emissions between the countries is similar to that seen for CFC-11 in panel (a), except for the Benelux which emitted less than the UK, Italy and Germany, though more than Ireland and Switzerland, whose emissions are relatively low (<0.05 Gg). France has the greatest estimated emissions, but shows a strong decline of 62% between 2008 and 2021. The decline in CFC-11 in France for the same period was 44%. Panel (d) shows that France has the highest emission per capita, followed by Italy and Ireland.

Figure 5 panel (e) shows that the decline in $CCl_4$ emissions split by countries is less clear than the WEU decline, Figure 3 panel (e). The dominant emission of $CCl_4$ in the WEU region comes from southern France with a sustained decrease from 2017 to 2021, but France's 2021 emissions are still more than double other WEU countries's emissions. France's per capita emissions are also clearly the largest (panel (f)). Unlike CFC-11 and CFC-12, where emissions are mainly from the remaining bank, there is less of a link between $CCl_4$ emissions and population. Emissions of this gas tend to arise from industrial processes resulting in a very different spatial distribution for $CCl_4$ compared to CFC-11 and CFC-12 (Figure 4). Fugitive emissions from the chlor-alkali industry are indicated by Graziosi et al. (2016) as potential source regions of $CCl_4$, which fits with the location in the south of France highlighted as the largest source in this study, and is in broad agreement the distribution of emissions modelled in Graziosi et al. (2016). Graziosi et al. (2016), however, found 2006-2014 emissions in the Benelux region of a similar magnitude to those in the south of France, whereas this study (2013-2021 Figure 4 panel (e)) has found much lower emissions in Benelux. Emissions from this region may have decreased in more recent years, though Figure 5 panel (e)) only shows a modest decline of $CCl_4$ from 2008 to 2014 for Benelux.





**Figure 5.** Panels (a), (c) and (e) show estimated emissions from 2008 to 2021 of the four model average by country, with 1 standard deviation uncertainty (shading), for CFC-11, CFC-12 and CCl$_4$, respectively. Panels (b), (d) and (f) show the emission per country (and average for Western Europe, WEU) in Mg per year per million of population (CIESIN, 2018) over the period 2008-2021, for CFC-11, CFC-12 and CCl$_4$, respectively.





The emissions from the CFC-11 banks arise due to leakage from foams in buildings, the release during the destruction of foams and from landfill. This study shows on average a 3.5 (2.7-4.8)% per year decrease in CFC-11 emissions between 2008 and 2021 and a 7.7 (6.3-8.0)% decrease per year in CFC-12 over the same period. We have calculated a bank release rate for CFC-11 of 3.4 (2.6-4.5) % (calculation detailed in supplement) which aligns with the bank release rate range of 1.5-4.2% reported by TEAP (2019). We have estimated a CFC-12 bank release rate of 7.7 (6.3-8.1)%. Based on the four model average,

for 2021 we calculate a CFC-11 bank of 55 (41-75) Gg and a CFC-12 bank of 10 (7-12) Gg for WEU. This is a small fraction (3.8 (2.8-5.2) % of what is estimated globally TEAP (2019) for CFC-11.

The rate of decline for CCl$_4$ is more complicated due to its permitted use as a chemical feedstock and consequent uncertainty around potential emissions from ongoing industrial processes. CCl$_4$ emissions in this study are decreasing on average by 4.4 (2.6-6.4)% per year across WEU, which is lower than the average 6.9% decrease (2006-2014) reported over a larger European

domain by Graziosi et al. (2016).

We can only speculate why Benelux (Belgium, Luxembourg, and the Netherlands) and north-east of France show enhanced CFC-11 emissions (Figure 4). There is a significant chlorine chemical industry in the region and it contains Europe's largest ports. Possibly, historical banks are higher in this region. When the emissions are scaled by population for the different countries, (Figure 5 panel (b)) the emissions from France and Benelux are similar, 168.3 ± 40.0 and 136.3 ± 50.0 Mg per million

people and much higher than that of the UK, Germany and Italy (69.0 ± 14.5, 73.7 ± 32.0 and 80.4 ± 39.0 Mg per million people respectively), which are countries with similar sized populations to France. Ireland's small relative population (approximately 8% of that of France in 2021) makes the result less comparable. The emission of CFC-11 across WEU for the period 2008-2021 shown in Figure 5 equates to an annual emission of 2.4 ± 0.4 Gg for a population of 335 million people (2021), which can be compared with emissions in the USA, with a population of 330 million people, of 5.6 ± 1.2 Gg (Hu et al., 2022),

indicating that US per capita emission are nearly twice those in WEU (on average) and comparable to those in France.

To try to understand the spatial distribution of CFC-11 in Europe it is useful to look at the distribution of the related chemicals, CFC-12 and CCl$_4$ (Figure 4). The distribution estimated from the average of the four inverse systems shows a slight elevation of CFC-12 emissions in the same area as seen for CFC-11. The estimated emissions of CCl$_4$ do show enhanced emissions from the same region, however the spatial distribution is dominated much more by the elevated emissions in the

south east of France. Together with the estimated WEU CFC-11 bank release rate discussed above, the lack of correlation in the spatial distribution of CFC-11 and CCl$_4$ suggests that the CFC-11 emissions are unlikely to be due to fugitive leaks from the chemical industry as it is reasonable to assume that these compounds would be co-emitted. Further research is required to get a better understanding of the geographical spread of CFC-11 use historically, as well as past and present locations for decommissioning and subsequent destruction of CFC-11 containing materials. A more comprehensive set of observations

closer to the Benelux and northern France geographical region, would also potentially shed further light on this region's elevated emissions.



# 4 Conclusions

In this study we have estimated the Western European emissions of CFC-11, CFC-12 and CCl$_4$ using four different inverse modelling systems and data from four atmospheric measurement sites for the period 2008 to 2021. This has allowed us to look at the spatial distribution of the emissions over this region and to calculate the recent trends in emission for each of these gases.

Over WEU we found that the average CFC-11 emission over the period 2008-2021 was $2.4 \pm 0.4$ Gg yr$^{-1}$ decreasing at a rate of 3.5 (2.7-4.8)% yr$^{-1}$. CFC-12 emissions were $1.3 \pm 0.3$ Gg yr$^{-1}$ decreasing at a rate of 7.7 (6.3-8.0)% yr$^{-1}$, and CCl$_4$ emissions were $0.9 \pm 0.2$ Gg yr$^{-1}$ decreasing at a rate of 4.4 (2.6-6.4)% yr$^{-1}$. Assuming that all emissions come from banks, we thus estimate bank release rates of 3.4 (2.6-4.5)% and 7.7 (6.3-8.1)% for CFC-11 and CFC-12 respectively, consistent with TEAP (2019). We have estimated the 2021 WEU bank for each gas to be 55 (41-75) Gg for CFC-11 and 10 (7-12) Gg for CFC-12. We find the highest CFC-11 emissions during the 2013-2021 period in Benelux and the north east of France, with smaller co-located CFC-12 emissions. The highest CCl$_4$ emissions are found in the south-east of France. This emission area is consistent with known emissions from chlor-alkali plants and previous work by Graziosi et al. (2016). France has the highest emissions per capita for all three gases. Despite the regions of higher emissions of CFC-11 in France and Benelux, as the emissions are reducing at a rate consistent with a decline in the bank, we do not consider this to be indicative of unreported production or consumption. Instead, it is likely a reflection of historic use and population density in these regions. We thus conclude that CFC-11 emissions are declining as expected in Western Europe.

*Code and data availability.* Atmospheric measurement data from AGAGE stations are available from (http://agage.mit.edu/data/agage-data, last access: 28 September 2022). Data from the Tacolneston observatory are available from the Centre for Environmental Data Analysis (CEDA) data archive: https://catalogue.ceda.ac.uk/ uuid/a18f43456c364789aac726ed365e41d1 (last access: 28 September 2022). The atmospheric observations from May 2020 are provisional. NAME and InTEM are available for research use and subject to licence, please contact the corresponding author. The Bristol MCMC inversion code is available at https://github.com/ACRG-Bristol/acrg (last access: 28 September 2022) or https://doi.org/10.5281/zenodo.6834888, (Rigby et al., 2022).

*Author contributions.* AJM and ALR ran the InTEM inverse model. SH ran the Empa FLEXINVERT system. FG ran the Urbino FLEXINVERT system. LMW ran the Bristol-MCMC system. Measurement data were collected by SOD, DY, JP, TGS, PGS, MHV, SR, JA, MM, JM, PKS, RFW, and KS. CMH produced and maintained the gravimetric SIO-05 calibration scales for these gases. ALR, AJM, SH, FG, LMW, ALG, JM and MR coauthors wrote the manuscript.

*Competing interests.* The authors declare that they have no competing interests.



*Acknowledgements.* UK Department of Business, Energy and Industrial Strategy (BEIS) (Contract number: TRN 1537/06/2018). ALR and
380 AJM were supported by the Met Office Hadley Centre Climate Programme funded by BEIS and Defra. The operation and calibration of
the global AGAGE measurement network are supported by NASA's Upper Atmosphere Research Program through grants NAG5-12669,
NNX07AE89G, NNX11AF17G, and NNX16AC98G (to MIT) and NNX07AE87G, NNX07AF09G, NNX11AF15G, and NNX11AF16G (to
SIO). Financial support for the Jungfraujoch measurements is acknowledged from the Swiss national program CLIMGAS-CH (Swiss Federal
Office for the Environment, FOEN) and from ICOS-CH (Integrated Carbon Observation System Research Infrastructure). Support for the
385 Jungfraujoch station was provided by International Foundation High Altitude Research Stations Jungfraujoch and Gornergrat (HFSJG). The
O. Vittori station (CMN) is supported by the National Research Council of Italy. LMW received funding from the European Union's Horizon
2020 research and innovation programme under the Marie Skłodowska-Curie grant agreement No 101030750. MR, ALG and LMW were
partially supported by NERC grants NE/S004211/1 and NE/V002996/1.



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
