# Peer review of "Western European emission estimates of CFC-11, CFC-12 and CCl4 derived from atmospheric measurements 2008 to 2021"

_EGUsphere, 2023_

## Referee Comment (RC1)

Comments to the manuscript (Lambert Kuijpers, lambert.kuijpers@kpnmail.nl)
**CFC-11 emissions are declining as expected in Western Europe**
**MS No.: egusphere-2023-40, by Redington, A. et al.**

**Overall quality**

Reading through the manuscript, there are some interesting observations, however, I am not sure whether the title (and the last sentence in the conclusions) really covers what is in the manuscript. The manuscript is not really aiming at specific CFC-11 emissions that are declining in Western Europe, but is gives an overview of the trends of the emissions for three chemicals (CFC-11, -12 and CTC) from Western Europe during the last 5-12 years, so to say. From the observations one concludes that emissions are declining, and identifies specific "hotspots" where there have been relatively (not excessively) large emissions in recent years. However, a conclusion that CFC-11 emissions are declining "as expected" has no scientific basis, since there have no patterns been published during the last 10 years forecasting these precise developments. Of course, anybody would logically expect that the emissions of CFC-11, CFC-12, and CTC would have decreased in Western Europe during the last decade or so. The manuscript addresses scientific questions, but does not present novel concepts and ideas, it presents some new info on tools and new data, of course.

In the abstract it says "The motivation for this work was to assess the emissions of CFC-11 and the associated gases, CFC-12 and $CCl_4$, from Western Europe". OK, clear, that has been done in the manuscript, and these data are now available. After an introduction, and a long description of measurement methods, then an analysis of emissions patterns, one concludes emission decreases, with certain differences, and with some regions where there have been (temporarily) higher emissions of one of the three chemicals. This without too much of an explanation why, while there are some "rough" assumptions mentioned. The conclusions mention again the emissions decrease of the three gases, mention some spots for higher emissions, and then say that "Despite the regions of higher emissions of CFC-11 in France and Benelux, as the emissions are reducing at a rate consistent with a decline in the bank, we do not consider this to be indicative of unreported production or consumption …. [..] ..  We thus conclude that CFC-11 emissions are declining as expected in Western Europe". Where the conclusions may adequately summarize what has been measured, with some special hotspots, the "we thus conclude" sentence is not good at all for me, and it is no conclusion in fact.

The overall presentation is well structured, the language is fluent and precise, and symbols, and units are correctly defined. However, my major question remains: will the reader get any important information here, that confirms other measurements, or that shows that unexpected things are happening? I do not think so, that implies that after reading the manuscript, one is inclined to say: "so what?". And that should not be result of a scientific paper, in my opinion. So, this may require changing the overall set-up of the manuscript, so it has more the character of new findings in a certain perspective, if at all.

**General comments**

Let me give some further impressions, and after that, a number of more detailed comments.

Abstract:

A lot of detailed information, also on other CFC-11 measurements in China, which are not relevant, or maybe confusing (line 4-8), not needed. There is lot of info on the regional measurements, the precision, etc. After that, there are a number of more detailed comments, which could be shortened, some info on CFC-12 and CTC is missing, and a much less strong statement (as expected, compared to the conclusions) is made on CFC-11 (why only on CFC-11, that is not clear): "Our estimated decline in 20 emissions of CFC-11 is consistent with a Western European bank release rate of 3.4 (2.6-4.5)%, which is in the upper half of the published range ….". Is that it (?), is that the conclusion, which is in fact *quite different* from the title and the last sentence in the conclusions.

**Introduction:**
A lot of information is included here. A certain amount has been given many times in many publications during the last 20-30 years, and some of it is not really correct here. The paragraphs on CFC-11 issues in China, and also the issues are USA and Australia are not relevant. The relevancy of that material for this article is not really elaborated upon here. It could be shortened, and it would imply that one would have to focus much more on the European issue. There are numbers given, yes, but the overall framework why and how to do this, that is not made clear.

**Methods:**
I cannot comment. But it will be a long text for the reader to read after the introduction, however, I understand that certain issues are coming back under results and discussions (and this needs detailed explanation here).

**Results:**
Lines 248-320 give all kind of results (also for regions), on the three chemicals. It is all good, but as mentioned the CFC-12 and CCl4 do not make it to the end. That means, there is no conclusion that there are certain relations, why things happen to one, or the other in comparison to others.

Lines 320-351 give various types of information. It first tends to go to bank releases from CFC-11, then starts to say something about bank releases from CFC-12, mentions that one should look at combined patterns, mentions regional issues. It may all be important, but I get lost what one actually wants to derive…… One sentence mentions: "We can only speculate why Benelux (Belgium, Luxembourg, and the Netherlands) and north-east of France show enhanced CFC-11 emissions (Figure 4). There is a significant chlorine chemical industry in the region and it contains Europe's largest ports. Possibly, historical banks are higher in this region". I cannot support such a statement, without further explanation. Are banks related to the presence of the chemical industry in the region? This is not really possible, or the real thing. This statement would really need to be further analyzed, so it would give some good information (e.g., one can also mention that Germany, Netherlands, Luxemburg etc. have had the most thorough program during 1994-2020 to take CFC-11 out of the foam and condense it as a liquid for destruction or for other purposes where it would not leak out (Becker, RAL, LUX, still coordinates the program. [Christoph.Becker@ral-online.org](mailto:Christoph.Becker@ral-online.org))).

**Conclusions:**
The authors give proper credit to related work and they more or less indicate their own new contribution. Where it concerns the references, no comments, the supplementary material seems OK.

There are no real conclusions in my opinion. Where the scientific methods and assumptions are valid and clear, the results are sufficient to support the interpretations, with some concluding remarks, but there is no real conclusion. At least not the sort of conclusion I want to see from this draft paper "We thus conclude that CFC-11 emissions are declining as expected in Western Europe".

**Specific issues**

A few line comments:

3        Dispersive use is to some degree correct, but it is just all uses (without feedstock), UNEP will never use dispersive uses in data

5        refrigerators, not clear, for CFC-11, yes. CFC-12 was another issue, mainly in mobile AC

5-7     China emissions do not belong here, confuse the picture, or it should JUST be mentioned that they can be excluded here

9        True, Europe phased out in 1995, however, the MP had a phaseout in 1996

28      use is not controlled

30      as in line 9

33      CFC-12 was used in MOBILE air conditioning (not much in large AC units) … it was used in refrigeration (domestic and retail, in retail small to large units were on R-502 and HCFC-22), not too much in foams

35      feedstocks are not exempted, feedstocks are allowed, they only need to be reported

38      emissions do not decrease as a result of the MP, that is a side effect, the MP is a consumption control mechanism

35-40  confusing as it concerns banks

43-45  Any TEAP estimate is not giving emissions; banks sizes are also reported by others (see the Lickley publications)

45-67  I do not think this is needed here the way it is now, it is not related to this paper (maybe delete, at least shorten substantially)

---

## Author Comment (AC1)

**Review ACP-2023-40**

We thank both the reviewers for their useful comments and suggestions which have helped to improve the manuscript. The reviewer's comments are dealt with in turn and are listed below in **blue**. Our responses are in **black** text and substantial changes that have been added to the manuscript are shown in italic in **green**. The revised abstract and Figures 1 and 4 are shown at the end of the document.

**First Review**

**Overall quality**

Reading through the manuscript, there are some interesting observations, however, I am not sure whether the title (and the last sentence in the conclusions) really covers what is in the manuscript. The manuscript is not really aiming at specific CFC-11 emissions that are declining in Western Europe, but is gives an overview of the trends of the emissions for three chemicals (CFC-11, -12 and CTC) from Western Europe during the last 5-12 years, so to say.

We have changed to the title of the paper to better reflect the content of the paper as suggested.

**Western European emission estimates of CFC-11, CFC-12 and CCl4 derived from atmospheric measurements 2008 to 2021**

From the observations one concludes that emissions are declining, and identifies specific "hotspots" where there have been relatively (not excessively) large emissions in recent years. However, a conclusion that CFC-11 emissions are declining "as expected" has no scientific basis, since there have no patterns been published during the last 10 years forecasting these precise developments. Of course, anybody would logically expect that the emissions of CFC-11, CFC-12, and CTC would have decreased in Western Europe during the last decade or so.

**We accept the reviewer's arguments and have removed 'as expected' throughout the paper.**

The manuscript addresses scientific questions, but does not present novel concepts and ideas, it presents some new info on tools and new data, of course. In the abstract it says "The motivation for this work was to assess the emissions of CFC-11 and the associated gases, CFC-12 and CCl4, from Western Europe". OK, clear, that has been done in the manuscript, and these data are now available. After an introduction, and a long description of measurement methods, then an analysis of emissions patterns, one concludes emission decreases, with certain differences, and with some regions where there have been (temporarily) higher emissions of one of the three chemicals. This without too much of an explanation why, while there are some "rough" assumptions mentioned. The conclusions mention again the emissions decrease of the three gases, mention some spots for higher emissions, and then say that "Despite the regions of higher emissions of CFC-11 in France and Benelux, as the emissions are reducing at a rate consistent with a decline in the bank, we do not consider this to be indicative of unreported production or consumption .... [..] .. We thus conclude that CFC-11 emissions are declining as expected in Western Europe". Where the conclusions may adequately summarize what has been measured, with some special hotspots, the "we thus conclude" sentence is not good at all for me, and it is no conclusion in fact.

**We have changed the last line of the conclusions to:**

This study concludes that emissions of CFC-11, CFC-12 and CCl4 have all declined from 2008 to 2021 in Western Europe. Therefore, this study finds no evidence that Western European emissions

**contributed to the unexplained part of the global increase in atmospheric concentrations of CFC-11 observed in the last decade.**

The overall presentation is well structured, the language is fluent and precise, and symbols, and units are correctly defined. However, my major question remains: will the reader get any important information here, that confirms other measurements, or that shows that unexpected things are happening? I do not think so, that implies that after reading the manuscript, one is inclined to say: "so what?". And that should not be result of a scientific paper, in my opinion. So, this may require changing the overall set-up of the manuscript, so it has more the character of new findings in a certain perspective, if at all.

We disagree with the reviewer on this point. There have been no recent published emission estimates of these three gases over Western Europe. However, there have been recent emission estimates from East Asia (Rigby et al 2019, Park et al 2021) that have shown unexpected emissions of CFC-11 in the last decade. These regional East Asian emissions do not explain all of the global increase in concentration, and therefore we believe that it is important to rule out other geographical areas where these emissions may have originated. The fact that there is a decline in emissions of these three gases over the last decade in Western Europe is therefore significant, and we believe, useful to the scientific community and international policy makers. The Editor should also note that the interpretations of these measurements are new, and there are no existing emission estimates of these gases in Western Europe, and therefore we are not confirming 'other' results. We believe that the absence of 'unexpected results' should not preclude publication given the global interest in emissions of these three gases.

**General comments**

Let me give some further impressions, and after that, a number of more detailed comments.

**Abstract: A lot of detailed information, also on other CFC-11 measurements in China, which are not relevant, or maybe confusing (line 4-8), not needed.**

We have edited the abstract, removing specific reference to East Asia but we felt it was important to explain that the unexpected global emissions in CFC-11 in the last decade have not been fully accounted for, hence in part our motivation for this work:

After 2010, emissions of CFC-11 and CFC-12 should therefore mostly originate from existing banks (e.g. from foams, mobile air conditioning units and refrigerators), however evidence has emerged of an increase in global emissions of CFC-11 in the last decade, some of which has not been fully accounted for. The motivation for this work was to assess the emissions of CFC-11, CFC-12 and CCl4, from Western Europe.

There is lot of info on the regional measurements, the precision, etc. After that, there are a number of more detailed comments, which could be shortened, some info on CFC-12 and CTC is missing, and a much less strong statement (as expected, compared to the conclusions) is made on CFC-11 (why only on CFC-11, that is not clear): "Our estimated decline in emissions of CFC-11 is consistent with a Western European bank release rate of 3.4 (2.6-4.5)%, which is in the upper half of the published range ....". Is that it (?), is that the conclusion, which is in fact quite different from the title and the last sentence in the conclusions.

The section 'which is in the upper half of the published range' has been removed from the abstract and more detail on the ranges from the TEAP report have been included in the discussion section. We have calculated a bank release rate for CFC-11 of 3.4 (2.6-4.5) % (calculation detailed in supplement) which aligns with the upper end of the bank release rate ranges of 1.5-2.5% (bottom up analysis) and 3.5-4.2% (derived from atmospheric measurements) summarised in the TEAP 2019 report, Appendix 6 sections A6.2 and A6.3 respectively.

**The concluding sentence of the abstract has been changed to the following:**

This study concludes that emissions of CFC-11, CFC-12 and CCl4 have all declined from 2008 to 2021 in Western Europe. Therefore, this study finds no evidence that Western European emissions contributed to the unexplained part of the global increase in atmospheric concentrations of CFC-11 observed in the last decade.

Introduction: A lot of information is included here. A certain amount has been given many times in many publications during the last 20-30 years, and some of it is not really correct here. The paragraphs on CFC-11 issues in China, and also the issues are USA and Australia are not relevant. The relevancy of that material for this article is not really elaborated upon here. It could be shortened, and it would imply that one would have to focus much more on the European issue. There are numbers given, yes, but the overall framework why and how to do this, that is not made clear.

We have to disagree and feel that it is important to summarise here both the CFC-11 work in China and studies from other regions for the three gases investigated, to give the reader the necessary background to understand the context of this study.

Methods: I cannot comment. But it will be a long text for the reader to read after the introduction, however, I understand that certain issues are coming back under results and discussions (and this needs detailed explanation here).

Results: Lines 248-320 give all kind of results (also for regions), on the three chemicals. It is all good, but as mentioned the CFC-12 and CCl4 do not make it to the end. That means, there is no conclusion that there are certain relations, why things happen to one, or the other in comparison to others.

As the reviewer states the text contains results and discussion for all three gases studied (lines 247-330), before focussing more on the spatial distribution of CFC-11 in the final paragraphs (331-350) with reference to CFC-12 and CCl4 as appropriate, which we feel is consistent with our motivation to explore the possible contribution of European emissions of CFC-11 to the unexplained global increase.

Lines 320-351 give various types of information. It first tends to go to bank releases from CFC-11, then starts to say something about bank releases from CFC-12, mentions that one should look at combined patterns, mentions regional issues. It may all be important, but I get lost what one actually wants to derive...... One sentence mentions: "We can only speculate why Benelux (Belgium, Luxembourg, and the Netherlands) and north-east of France show enhanced CFC-11 emissions (Figure 4). There is a significant chlorine chemical industry in the region and it contains Europe's largest ports. Possibly, historical banks are higher in this region". I cannot support such a statement, without further explanation. Are banks related to the presence of the chemical industry in the region? This is not really possible, or the real thing. This statement would really need to be further analyzed, so it would give some good information (e.g., one can also mention that Germany, Netherlands, Luxemburg etc. have had the most thorough program during 1994-2020 to take CFC-11 out of the foam and condense it as a liquid for destruction or for other purposes where it would not leak out (Becker, RAL, LUX, still coordinates the program. Christoph.Becker@ral-online.org)).

**The text has been amended to:**

**We do not know why Benelux (Belgium, Luxembourg, and the Netherlands) and north-east of France show enhanced CFC-11 emissions (Figure 4). Possibly, historical banks are higher in this region due to high population density.**

It is beyond the scope of this paper to analyse which sources the emissions in this region may have originated from as this is a modelling study based on atmospheric data – i.e. purely a top down study not a bottom up analysis, so we have removed any speculation on possible sources.

Conclusions: The authors give proper credit to related work and they more or less indicate their own new contribution. Where it concerns the references, no comments, the supplementary material seems OK. There are no real conclusions in my opinion. Where the scientific methods and assumptions are valid and clear, the results are sufficient to support the interpretations, with some concluding remarks, but there is no real conclusion. At least not the sort of conclusion I want to see from this draft paper "We thus conclude that CFC-11 emissions are declining as expected in Western Europe".

We have re-worked the conclusion to tie in better with the motivation for the work, which was to assess the current status of these gases in Western Europe to rule out any possible contribution to the yet fully explained increase in global CFC-11 in the last decade.

We conclude that emissions of CFC-11, CFC-12 and CCl4 have all declined from 2008 to 2021 in Western Europe and hence that it is unlikely that this region has contributed to the unexplained part of the global increase in CFC-11 seen in the last decade.

**Specific issues**

A few line comments:

- 3 Dispersive use is to some degree correct, but it is just all uses (without feedstock), UNEP will never use dispersive uses in data corrected
- 5 refrigerators, not clear, for CFC-11, yes. CFC-12 was another issue, mainly in mobile AC text amended to include mobile air conditioning units
- 5-7 China emissions do not belong here, confuse the picture, or it should JUST be mentioned that they can be excluded here We have edited the abstract, removing specific reference to East Asia see earlier comment and revised text above
- 9 True, Europe phased out in 1995, however, the MP had a phaseout in 1996 Thank you for this clarification, corrected.
- 28 use is not controlled text amended
- 30 as in line 9 corrected
- 33 CFC-12 was used in MOBILE air conditioning (not much in large AC units) ... it was used in refrigeration (domestic and retail, in retail small to large units were on R-502 and HCFC-22), not too much in foams text amended to correct to mobile a/c units
- 35 feedstocks are not exempted, feedstocks are allowed, they only need to be reported Agreed, this is what we have said:

Production and consumption of CCl4 has been banned under the MP since 2010, with the exception of use as a chemical feedstock. Feedstocks are permitted to be produced under the assumption that the majority of CCl4 made, is fully converted into the target chemical, recycled or destroyed.

- 38 emissions do not decrease as a result of the MP, that is a side effect, the MP is a consumption control mechanism Text amended: Globally, emissions of CFC-11 and CFC-12 have been decreasing as a result of the consumption controls imposed by the MP.
- 35-40 confusing as it concerns banks See text for next point below
- 43-45 Any TEAP estimate is not giving emissions; banks sizes are also reported by others (see the Lickley publications) We are intending to talk about banks here and have added that almost all current emissions are from banks. Thank you for pointing out these additional references, Lickley 2020 has been included in the text.
- 45-67 I do not think this is needed here the way it is now, it is not related to this paper (maybe delete, at least shorten substantially) We disagree, see comments made above on Page 2.

**Second Review**

**Review of "CFC-11 emissions are declining as expected in Western Europe "**

This manuscript is well-written and the science discussed is very relevant to global atmospheric emissions, ozone loss and climate change. It also shows the importance of policy decisions to cut global emissions.

**Major:**

1. I am not sure what you expect and whether you have seen those results from the analysis. This is not discussed anywhere in the text. Also, if you look at the results, there are some years with high emissions. This is also expected? Figure 5, for Benelux, there is a slight positive trend? Therefore, it is better to avoid "expected" from the title. I do agree that there is a significant reduction in emissions.

We agree with this comment and have changed the title, abstract and conclusions to avoid the use of 'expected'. The changes are detailed above as they were also made by the first reviewer.

Some years do have higher modelled emissions than others - there is some year to year variability, however we would caution reading too much into this as it may be due to errors in the estimates made by the models. Therefore, we focus on the trend over time rather than year to year variation in this work. With Benelux we do not think there is a significant positive trend and it will require further years of data to confirm this.

We have added the following text to the results and discussion section to clarify our thoughts on year to year variability for the reader:

**Some year to year variability can be seen in the results, however we would caution reading too much into this as it may be due to errors in the estimates made by the models. Therefore,**

we focus on the trend over time, of the four model average, rather than year to year variation in this work.

2. You have discussed CFC-11, CFC-12 and CCl4 emissions, but the title says just CFC-11?

We have changed the title to better reflect the content of the paper

Western European emission estimates of CFC-11, CFC-12 and CCl4 derived from atmospheric measurements 2008 to 2021

3. "France also has the highest emissions per capita", is it also holds good for the other countries mentioned here?

Figure 5 (b,d,f) shows the emission per million of population by country for each of the three gases. France shows the highest values, although the uncertainties are quite large.

**Minor:**

Line 9: ", and " comma has been moved

L10: from 2008 to 2021 corrected

L14: any reason for these highest regional emissions or are you talking about the slow decline there? This has been re-worded to aid clarity.

Even though the emissions are declining throughout the period, the area including Northern France, Belgium, the Netherlands and Luxembourg show consistently elevated emissions of CFC-11 compared with the surrounding regions.

Abstract: The first 10 lines can be shortened We have removed the specific reference to East Asian emissions.

L40: all active banks are done

L49: Is it bank or banks? be consistent in using this corrected to banks

L57: from 2012 to 2017 to understand done

L58: In fact, they speculated that, and were not reporting agree

L106: ", respectively" done

L261: any reason for this particular increase in specific years, e.g. 1995-1996? In Figure 3 (f) it is not possible to speculate whether the peak in CCl4 from 1995-96 is real or not. CCl4 is used as a chemical feedstock for production of a variety of chemicals, and whilst it is possible that this is a real increase in emissions it should be noted that at this point in time (1995/6) we only have one set of measurement data from Mace Head in Ireland and this is the emission result from 1 model, hence the high model uncertainty.

L289: no single model particularly stands out? done

L302: between 2008 and 2021 done

L304: which emitted less than that of not changed - didn't seem to read well

L308: as shown in Figure 3 panel (e) done

Figure 5: Why Benelux shows an increase in the emission of CFC-11? We do not think there is a slight positive trend currently but it will require further years of data to confirm this.

L341: To understand done

Figure 1: font size is too small to read This figure has been remade with larger font size, included at the end of this document.

Figure 4: colour bar is very bad, as there is a repetition of the same (very similar) colour. Please change this colour bar. The colour bar has been changed, figure included below.

Figure 5: Any reason for the particular peaks? For instance, CFC-11 in 2010 in France and 2020 in Germany? Please also see response above to Major 1. Again we cannot speculate whether these are real signals or due to variability within the models as they fall within the model uncertainty. There are considerable year to year variations, hence why the paper is focussing on the trends over the time period 2008 to 2021 rather than shorter time periods.

**Abstract**

Production and consumption of CFC-11 (trichlorofluoromethane, CCl3F), CFC-12 (dichlorodifluoromethane, CCl2F2) and CCl4 (carbon tetrachloride) are controlled under the regulations of the Montreal Protocol and have been phased out globally since 2010. Only CCl4 is still widely produced under exemption as a chemical feedstock (non-dispersive use). After 2010, emissions of CFC-11 and CFC-12 should therefore mostly originate from existing banks (e.g. from foams, mobile air conditioning units and refrigerators), however evidence has emerged of an increase in global emissions of CFC-11 in the last decade, some of which has not been fully accounted for. The motivation for this work was to assess the emissions of CFC-11, CFC-12 and CCl4, from western *Europe. All countries in this region have been subject to the controls of the Montreal Protocol since* the late 1980s and, as non-Article-5 Parties, have been prohibited from producing CFCs and CCl4 for dispersive use since 1996. Four different inverse modelling systems are used to estimate emissions of these gases from 2008 to 2021 using data from four atmospheric measurement stations: Mace Head (Ireland), Jungfraujoch (Switzerland), Monte Cimone (Italy) and Tacolneston (UK). The average of the four model studies found that western European emissions of CFC-11, CFC-12 and CCl4 between 2008 and 2021 were declining at 3.5 (2.7-4.8)%, 7.7 (6.3-8.0)% and 4.4 (2.6-6.4)% yr-1 respectively. Even though the emissions are declining throughout the period, the area including Northern France, Belgium, the Netherlands and Luxembourg show consistently elevated emissions of CFC-11 compared with the surrounding regions. Emissions of CFC-12 were slightly elevated in the same region. CCl4 emissions were highest in the south of France. France had the highest emissions of all three gases over the period 2008-2021. Emissions from western Europe (2008-2021) were on average 2.4 +/- 0.4 *Gg* (*CFC-11*), 1.3 +/- 0.3 *Gg* (*CFC-12*), 0.9 +/- 0.2 *Gg* (*CCl4*). Our estimated decline in emissions of *CFC-*11 is consistent with a western European bank release rate of 3.4 (2.6-4.5)%. This study concludes that emissions of CFC-11, CFC-12 and CCl4 have all declined from 2008 to 2021 in western Europe. Therefore, this study finds no evidence that western European emissions contributed to the unexplained part of the global increase in atmospheric concentrations of CFC-11 observed in the last decade.

Figure 4

---

## Referee Report (RR1)

**Review comments, 25 May 2023**

**Abstract**

Line 3: Only $CCl_4$ is still widely produced under exemption as a chemical feedstock (non-dispersive use)

Better is: … is still widely produced as a chemical feedstock.

Forget about exemptions, non-dispersive use, etc. The latter term is a non- MP term anyhow.

Parties are free to produce any controlled substance as feedstock, as long as they report production for domestic use, exports and imports

Line 4:  ….. originate from existing banks (e.g. from foams, mobile air conditioning units and refrigerators)

Question is …. Can one imagine that there are still MAC units emitting CFC-12 ????, maybe a few vintage cars. If it is not retrofitted.

The following is being mentioned in line 7: …… All countries in this region have been subject to the controls of the Montreal Protocol since the late 1980s and, as non-Article 5 **(delete the "-" in Article-5)** Parties, have been prohibited from producing CFCs and $CCl_4$ for dispersive use since 1996.

This is true. However, in fact, Europe (including the UK) decided to phase out (one year earlier) by 1995, so that would be the real date. One can also use the Montreal Protocol date, but then it should also be changed in the paper further down, where Europe has been mentioned.

Line 14: ….. should be "shows"

Line 15-16: Western Europe (2008-2021) …. Has Western Europe been defined in the paper ?

Good to pick it up now. It has not, and WEU should not be used for Western Europe.

BTW, Line 292 says that WEU is Western Europe

However,

The Western European Union (WEU) was the international organization and military alliance that succeeded the Western Union after the 1954 amendment of the 1948 Treaty of Brussels. The WEU implemented the Modified Brussels Treaty. **Wikipedia**

**1 Introduction**

Line 27: Their production, consumption and use are controlled through the Montreal Protocol
,,,,
The Montreal Protocol does not control the use (and it does not control emissions)

Line 28: For non-Annex 5 parties

In climate, we have Annex I and non-Annex I, here it should be non-Article 5 parties (or Parties as in line 7-8

Line 29: should be Article 5 Parties (no dash)

"Western Europe" needs to be clear, also in the introduction (see also above), ……and when the phase-out of Annex A substances occurred

Line 30: ……..CFC-11 was mainly used in aerosol spray cans, as a solvent and as an agent for blowing foams into buildings and consumer products; CFC-12 was mainly used in refrigerators, mobile air conditioning units and as a foam blowing agent.

Does this apply to developed and/or developing countries ?

Line 31-32: $CCl_4$ was used historically as a solvent and also as a feedstock to produce other chemicals, predominantly CFC-11 and CFC-12. Production and consumption of $CCl_4$ has been banned under the MP since 2010.

Does this apply to the developing countries (or Article 5 Parties) ?

Line 34: …..with the exception of use as a chemical feedstock. Feedstocks are permitted to be produced under the assumption that the majority of $CCl_4$ made, is fully converted into the target chemical, recycled or destroyed……..

Please, leave the "Feedstocks are permitted…" sentence out, or just refer to the MP definition (I would not recommend) Line 36: "Globally, emissions of CFC-11 and CFC-12 have been decreasing as a result of the consumption controls imposed by the MP".

That is an interesting one, is it the production or consumption control ?.. The chicken or the egg ? I would use production controls, or both.

Sentence in lines 183-185, please check. The use of the references makes it a bit difficult to read.

Line 195: …..250%, I know this is normally used, but the standard is 250 %, in that way 250 stands out more

Line 244 shows the difficulty again with "Western Europe", one mentions here Spain and Poland, is Poland "Central Europe" ?

Line 256: "the phase-out in Europe during 1995"
To avoid confusion, it might be better to keep the MP date, 1/1/1996

Line 258: Is it necessarily "infrastructure" ?

Line 296: Benelux = B + NL + LUX, sometimes called Benelux countries, more often the Benelux countries. Up to authors, but the Benelux is used in line 350 !!

Line 284: France may leak some CCl4 from feedstock production, but there are some more countries in W-Europe that produce CCl4, with perhaps –small-- leakages (e.g., Germany)

. Line 348 says: …… from the chemical industry, as these amounts would be co-emitted …...
CCl4 is still used as feedstock in large quantities (but this doe not relate to CFC-11 and -12), so, leave co-emissions out here, or improve the sentence

**Conclusions**

Avoid western Europe and WEU, define something "good" in the Introduction

Line 356: "Over WEU" ??

Line 358: "all emissions from banks" ??, no emissions possible from CCl4 feedstock ?

Line 361, 364: maybe "the Benelux"

Lines 366-369 have two sentences that both use "this study". I think that the second sentence can do without "this study", maybe "Therefore, no evidence is found…"

---

## Author Response (AR2)

Response to Second Review – Reviewer 1

Thank you for your comments and corrections. These have all been dealt with except for line 4 where the MAC units were added in at the request of First review, reviewer 1, and have been left in.

*Line 3 amended*

*Line 4 MAC units were added at the request of original reviewer 1, and has been left in.*

*Europe from 1995 added to the introduction.*

*WEU has been replaced by W-Europe, and the definition has been moved to the introduction.*

*Annex corrected to Article*

*New paragraph to make it clear the uses apply to both groups.*

*CCl4 Montreal protocol date of 2010 was for all countries and non-Article 5 from 1996. This has been made clear in the text.*

*Feedstocks sentence removed as suggested.*

*Production added.*

*A space has been included before a '%'*

*"existing infrastructure" has been replaced with "existing products and buildings and in landfill"*

*Text amended to "the Benelux countries"*